# On the Use of Radar and Optical Satellite Imagery for the Monitoring of Flood Hazards on Heritage Sites in Southern Sinai, Egypt

Wael Attia [1], Dina Ragab [2], Atef M. Abdel-Hamid [3], Aly M. Marghani [4], Abdelaziz Elfadaly [1,*] and Rosa Lasaponara [5]

1 National Authority for Remote Sensing and Space Sciences, Cairo 1564, Egypt; wael.attia@narss.sci.eg
2 National Research Centre, Cairo 12622, Egypt; dinaabdelmoneim@u.boisestate.edu
3 Faculty of Arts, Cairo University, Cairo 12613, Egypt; atefoov@gmail.com
4 Faculty of Arts, Benha University, Benha 13511, Egypt; d.alykamel@hotmail.com
5 Italian National Research Council, C.da Santa Loja, Tito Scalo, 85050 Potenza, Italy; rosa.lasaponara@imaa.cnr.it
* Correspondence: abdelaziz.elfadaly@narss.sci.eg

**Abstract:** This study focuses on the use of radar and optical satellite imagery for flood hazard mapping and monitoring around the archaeological sites of the Wadi Baba area, situated at Sinai (Egypt) and well known for its heritage treasures belonging to diverse historical periods and civilizations from the Pharaonic, Nabateans, Christian, and Islamic eras. Although this area is located in an arid to semi-arid climatic region, it is intermittently flooded due to torrential rainstorms. To assess the amount of rainfall expected and its impacts on heritage sites, satellite Sentinel-1 (C-Band) and Tropical Rainfall Monitoring Mission (TRMM) data were jointly used with measurements from meteorological stations and the Digital Elevation Model (DEM) from Shuttle Radar Topography Mission (SRTM). Envi5.1, ArcGIS 10.4.1, Snap 6.0, and the GEE platform were used to process optical and radar data, which were further analysed using the ArcHydro model. In this study, the TRMM accumulated rainfall data acquired on 17 January 2010, Sentinel-1 radar images between 2017 and 2019, and Sentinel-1 data captured from 1 to 30 March 2020 processed by GEE platform were chosen to assess the effects of flood events on the archaeological sites in the study area. The results indicated that the study area is exposed to flood risk that significantly threatens these heritage sites. Based on that, mitigation strategies were devised and recommended to mitigate the flood hazard impact around the archaeological areas.

**Keywords:** flood hazards; Sentinel-1 data; cultural heritage management; archaeological sites' sustainability

## 1. Introduction

Wadi Baba is considered an important heritage area that includes thousands of Malachite reduction furnace units dating mostly to the Old Kingdom [1]. Indeed, natural hazards (e.g., floods) have caused major loss of human lives and archaeological landscape damage [2]. Flash floods frequently occur in many regions of Egypt, with damages of 19.2 million euros and 648 deaths estimated for 1990–2006 [3]. Recent floods led to six deaths and devastated about 2000 houses, with the damages estimated at as much as 6.7 million euros [4]. More than 40 flash floods have been recorded in the El Arish watershed during the last 100 years [5]. In the Sinai peninsula, flash flooding episodes result from sudden, heavy, and short-duration rainfall, and represent a risk to the population and to archaeological sites [6].

Given that floods are difficult to control, a reliable map of flood-prone areas is critical for flood management [7,8]. Recently, satellite optical and radar imagery has been able

to provide information useful to assess the condition of heritage at risk [9]. GIS and remote sensing techniques have been widely used for risk assessment of natural threats to archaeological sites over large and inaccessible areas [10,11]. Space technology in general has made a great contribution to various aspects of flood disaster management (e.g., prevention, preparedness, and relief) [12,13]. Radar satellite data can provide real-time mapping of flood extent and rapid data acquisition during flood events [14,15]. Radar satellites facilitate both the continuous and regular monitoring of the volume of floods and the mapping of flood risk zones [16,17].

In this study, satellite optical, radar, and tropical rainfall monitoring mission (TRMM) data, along with in situ meteorological station measurements, were used for assessing flood hazards on the archaeological sites of Wadi Baba. The ArcHydro model is regarded as a reliable hydrological modeling tool, herein coupled with the elevation map (DEM) from the SRTM data [18–20]. This enabled us to extract and take into account the spatial heterogeneity of the watershed as well as the morphometric delineation of the river basin, necessary to set up an engineering model to define mitigation strategies and reduce the impact of floods on the archaeological sites of Wadi Baba.

### 1.1. The Study Area

The Baba basin is located in the Sinai Peninsula at the northeastern corner of Egypt, and its outlet is at the shoreline of the Suez Gulf. The basin extends between 33°11′20″ E and 33°44′21″ E and 28°51′30″ N and 29°12′50″ N and covers about 721 square km. The Baba basin is vulnerable to flash flood hazards and damage due to torrential rainstorms [21]. The Wadi also includes several priceless archaeological and cultural heritage sites (e.g., Wadi El-Lahian, Wadi El-Rakaiz, Turquoise Grotto, Kharig, and Serabit el-Khadim Temple) (Figure 1).

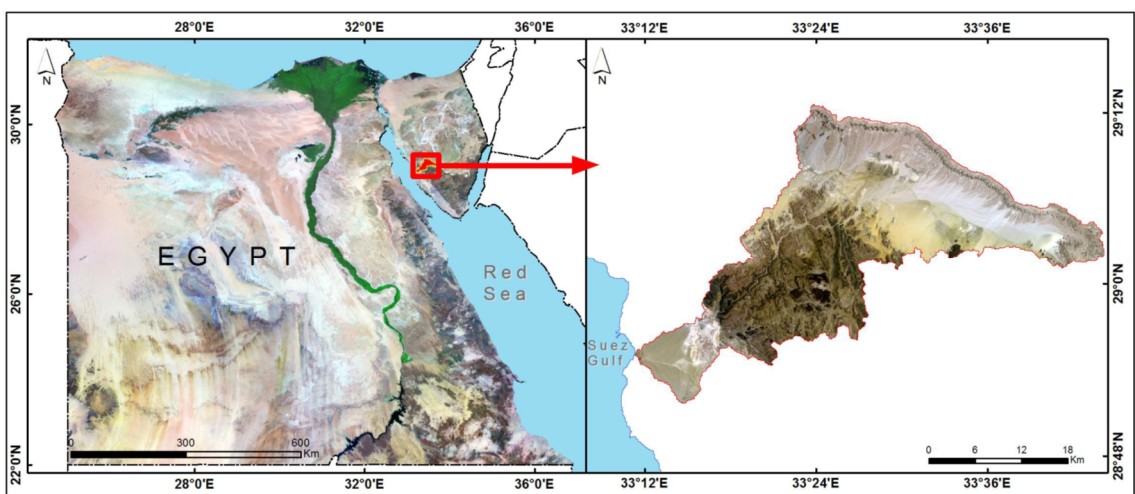

**Figure 1.** The location of the Baba basin and its boundary delineation from Landsat-7 DEM.

### 1.2. Problem Definition

In arid regions, such as the Sinai Peninsula in Egypt, heavy rainfall causes many flash flood events due to the convective cloud mechanisms and squall line, in addition to the low-intensity frontal rain that causes storm floods [22]. Due to the meteorological conditions of South Sinai, flood events are not frequent, but severe. In the study area, the affected regions include tourist locations, as well as residences of local inhabitants [23]. Therefore, it is necessary not only to protect these touristic sites' residence areas but also to achieve the maximum benefit from the runoff instead of wasting it in the Gulf of Suez [24] (Figure 2a–f).

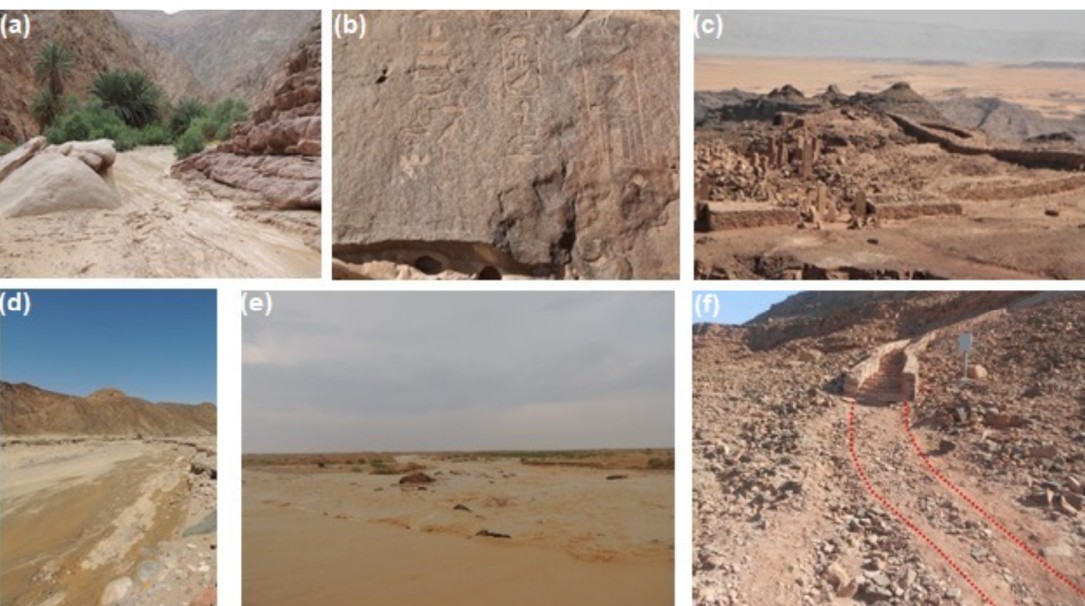

**Figure 2.** (**a**) The landscape and topographic view of the study area of Wadi Baba, (**b**) some of the inscriptions, (**c**) Serabit el-Khadim temple, (**d,e**) flooding event across Wadi Baba, (**f**) some of the deteriorated heritage sites in the study area due to the flood hazard in Wadi Baba.

## 2. Materials and Methods

### 2.1. Data Collection

In this study, SRTM (Shuttle Radar Topography Mission) and Sentinel-1 radar data (S1A 1 August 2017 and S1B 10 February 2019 and 1–30 March 2020) were used for detecting flooded regions. The Optical (spot4 and Landsat7 satellite imagery), TRMM (Tropical Rainfall Monitoring Mission), and rainfall data (depending on the observation station of Abu Zenima at southwestern Sinai) used to assess rainfall volume and devise a mitigation strategy are herein proposed to limit the impact of the flood risk on heritage sites.

### 2.2. Methods

#### 2.2.1. Optical Data Processing

In the current study, Envi5.1, ArcGIS 10.4.1, Snap 6.0 (the Sentinel application platform), and the GEE platform were used for data processing, technical analyses, and integration through applying some of the algorithm equations. Spot4 and Landsat7 optical satellite data were geometrically corrected, filtered, and also removed from their dropouts. These data were enhanced to improve the quality of the images. The data were geometrically corrected based on high-resolution data (Sentinel-2) having the spatial reference WGS_1984 with UTM_zone_36N. For increasing the layer class accuracy, the input and the results of the visual interpretation were analysed using ArcGIS, SNAP, and ENVI software.

#### 2.2.2. Radar Data Analysed Using SNAP Software

In this study, the two chosen SAR dates were analysed as follows; (i) orbit file refinement (apply orbit file), (ii) obtaining a subset of the whole image by setting the geographic coordinate values. After image subsetting, for getting a pixel value directly related to radar backscatter, radiometric correction was applied in SNAP software. After that, the image was calibrated (radiometric calibration), which is an essential step for quantitative use of Sentinel-1 data as the pixel values represent the true radar backscatter of the received surface object and the processing of Sentinel-1 data was dependent on the VV polarity. After the radiometric calibration, the multilook method was applied for Sentinel-1 images [25]. Furthermore, for reducing the speckle (speck filter), and at the same time preserving the radiometric and textural information in SAR images for enhancing the

visualization, some adaptive filters were applied. Since the obtained images have the same geometry as the sensor (terrain correction), it was necessary to change their projection into a geographic projection [26]. Also, the scenes were initially stacked and co-registered into a single reference (master) geometry [27]. In addition, the different polarizations observed by Sentinel-1a/b in the processing steps allow us to deduce more information about the flooded area, and with the co-polarization (VV) a stronger return over the flooded area is observed. Therefore, we create an RGB false-color composite to detect the differences between pre- and post-flood data as in Equation (1) [28]:

$$R: 〚20〛\,^* \log10(|VVpost|)$$
$$G: 〚20〛\,^* \log10(|VHpost|)$$
$$B: 〚20〛\,^* \log10(0.5^* |VVpre| + |VHpre|)).$$

(1)

SRTM data were used for the extraction of various topographic settings such as the slope (magnitudes and aspects) and hydrological parameters including fill, flow direction, watershed delineation, flow accumulation, stream networks, and flow length. After the basin boundary delineation, the extracted DEM of the basin went through a series of processes such as filing to remove small imperfections in the data; flow direction, where the D-8 method was used to get the direction of flow out of every cell; flow accumulation; drainage network; and flow length to get the longest length of the flow. The morphometric parameters of the basin were derived from the results of the digital elevation model analyses. Also, the evaluated morphometric parameters were grouped as relief, linear, or areal parameters. The watershed was subdivided into four subcatchments to examine the hydrological processes within them. The time of concentration was defined as the time the water takes to travel from the most remote point on the catchment to the outlet. In the same context, it depends on watershed properties such as slope, length, and area, according to KIRPICH/RAMSER in Equation (2) [29]:

$$Tc = 0.0195\,L\,0.77\,〚\quad〛\,^* S - 0.385$$

(2)

where $Tc$ = the time of concentration (min), $L$ = the length of the main Wadi (m), and $S$ = distance weighted channel slope (m/m).

The basin lag time (TLAG) is defined as the time between rainfall excess and runoff inside the area of the basin, which considers a critical parameter in the runoff routing models [30]. Indeed, the basin lag time was calculated according to Equation (3) [31]:

$$Tk = ML0.77\,S - 0.39,$$

(3)

where $Tk$ = the critical lag in hours; $L$ = the distance from the outlet to the most remote part of catchment along the path of flow in miles; $S$ = flow slope for $L$ in ft/ft; and $M$ = a constant to be evaluated (0.6 for the poorest pasture and desert vegetation) [32].

2.2.3. Radar Data Analysed Using the GEE Platform

Firstly, the location coordinates of the study area were detected using the following five points: [2.98657594471415, 28.720432209216987], [34.00281129627665, 28.720432209216987], [34.00281129627665, 29.269361991840636], [32.98657594471415, 29.269361991840636], and [32.98657594471415, 28.720432209216987]. Then, Sentinel-1 imagery (GRD_ C-band) was collected inside the GEE platform in the transmitter Receiver Polarisation 'VV'. The dates chosen for measurement during March 2020, based on the known rainfall-specified days, were divided based on the rainfall time into the two halves of the month, the first from 1 to 13 March and the second from 14 to 30 March. After that, the threshold-smoothed radar intensities were applied automatically inside the platform to identify the flooded areas. The SMOOTHING_RADIUS equaled 100 and the DIFF_UPPER_THRESHOLD equaled –3. Finally, the watershed throughout the study area was calculated by extracting the terrain

and slope of the study area using the internal Sentinel-1 radar satellite imagery by the pass 'DESCENDING' [33].

### 2.2.4. Estimating the Runoff Volume and Peak Flow Rate

In the study area, the major factors affecting the runoff volume and associated peak discharge were the rainfall duration and intensity, time of concentration, and soil types in the watershed. The rational equation for estimating the runoff volume is shown in Equation (4) [34]:

$$V = R * C * A * P$$
$$R = 1.05 - 0.0053 \, A,$$

$$(4)$$

where V = Surface runoff value, P = rainfall depth, C = the runoff coefficient, A = the basin area, and R = the reduction factor depending on the basin area.

On the other hand, the peak runoff (QR) is defined as the maximum flow that results from an individual storm. The rational formula method considers that the total drainage area is calculated as a single unit, the drainage area flows at the most downstream point only, and the rainfall is uniformly distributed through the drainage area. The peak flow rate is determined by Equation (5) [35]:

$$Q = CIA/360,$$

$$(5)$$

where Q = peak flow rate (maximum runoff, $m^3/s$), A = catchment area, I = rainfall intensity (mm/h), and C = the runoff coefficient, which can be computed with Equation (6) [36]:

$$C = (((D * \sqrt{((9.81 * P)))})/G)\,\hat{}\,(1 - S),$$

$$(6)$$

where C = runoff coefficient, P = the rainfall depth (mm), D = drainage density ($km/km^2$), S = the average land surface slope, and G = an integer value representing the geology of the surface. The inputted data, preprocessing, processing, postprocessing steps, and results of the study are shown in Figure 3.

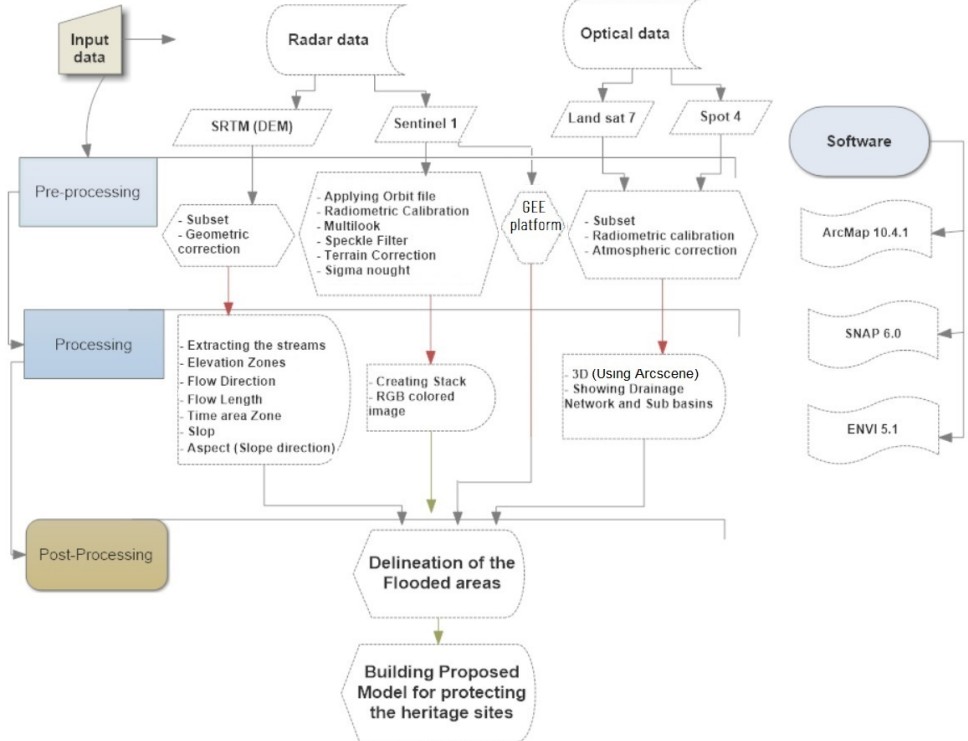

**Figure 3.** Flowchart showing the data collected and method (preprocessing, processing, and postprocessing steps).

## 3. Results and Discussion

### 3.1. Satellite Image Results

The per 3 h TRMM accumulated rainfall data acquired on 17 January 2010 were used to conceptualize the rainfall intensity of the basin (Figure 4a,b). Also, the drainage network, the stream orders, and slope direction were extracted from the DEM data and matched with Spot-4. Furthermore, the contour lines, stream orders, and water depth in the study area were extracted and calculated based on the DEM data. On the other hand, remote sensing data, especially radar data, became familiar tools used in detecting flood events around the world as a result of the accurate results, low costs, and continuous observation [37–39]. In this study, for adapting the implementation of Sentinel-1 data to view flood mapping, the following steps were applied. To improve processing throughput, image tiling was included. The next step focused on replacing the nonlocal means filter with a curvelet-based approach for enhancing speckle suppression along with the often curvilinear outlines of the flooded regions. Then, integrated Bayesian inferencing was used for the enhancement of the thresholding performance. For showing flooded areas, RGB composites were created from the two sets of Sentinel-1 data (S1A 1 August 2017 and S1B 10 February 2019) depending on the dual-pol S1 GRDH data; these results, in RGB-format images, show the look and feel of visual-band data. The approach transfers the co- and cross-pol signal into a simple bounce (polarized) with some value scattering and volume (depolarized) scattering, and a simple bounce with quite low-value scattering. These channels are assigned to (R) red, (G) green, and (B) blue. The convenience of the red, green, blue decomposition for the created flood map stems from the fact that the B channel is mostly linked with standing water, adding an easy-to-understand visual aid for responsible personnel (Figure 5).

The extracted hydro-shed data from the radar data were exported as images to the internal drive of a computer for both types of data collected from the GEE platform. The data scale was chosen in 10, the region was chosen in geometry, and the max pixel equaled 3E10. The outputs were converted from raw raster data to polygon layers for matching the extracted data with the rest of the optical data in an easier way. Finally, the threatened areas were detected according to the Sentinel-1 radar data from both chosen time periods (1–13 March and 14–30 March 2020). The results of this study showed that the archaeological sites were most affected by the flooding events. The archaeological sites of Serabit El-Khadim, Wadi Kharig, Wadi Nasab, Wadi Lahyan, El-Sehou, Wadi Shalaal, and Rakiz were affected by the flood events between 1 and 13 March 2020 that focused on the east and middle of the study area. On the other hand, the flood events were focused more on the west of the study area between 14 and 30 March had a major effect on built-up wells and roads on the coast of the Suez Canal (Figure 6).

### 3.2. Extracting Morphometric and Hydrologic Parameters

The watershed morphometric analysis is crucial for quantitative description [40]. The morphometric features provide information concerning the watershed formation and development because all geomorphic and hydrologic processes take place within the watershed [41]. However, the key objective of calculating these parameters was to extract the hydrologic ones. Therefore, 22 linear, relief, and areal morphometric parameters were computed as summarized in Table 1; some of these were, in turn, used for calculating the hydrological parameters of the four sub-basins using GIS techniques. Hydrological parameters are useful for flood management because these parameters can tell us (i) to what extent the watershed is sensitive to intensive storms, (ii) which watershed is more vulnerable to repeated flood events, and (iii) how much the runoff volume is across the basin. For instance, the hydrological parameters (i.e., time of concentration and lag time) for Wadi Naga Elfda are the highest, followed by Wadi Akhfi, Wadi Elshlal, and Wadi Elseih (Table 2). Thus, Wadi Naga Elfda is less sensitive to short-duration but extreme-intensity storms than the other sub-basins. In addition, Wadi Naga Elfda is less vulnerable to repeated flood episodes. Wadi Elseih is extremely vulnerable to repeated flood events; this vulnerability decreases at Wadi Elshlal and Wadi Akhfi. Wadi Akhfi produces the greatest

runoff volume (Figure 7) of approximately $190.4 \times 10^3$ m$^3$/h, representing about 37.8% of the total runoff volume of the study area, followed by Wadi Naga Elfda, Wadi Elseih, and Wadi Elshlal with discharge volumes of $183.5 \times 10^3$, $86.6 \times 10^3$, and $43.2 \times 10^3$ m$^3$/h, respectively. Wadi Akhfi has the largest peak flow rate among the four sub-basins; the maximum runoff values range between 54 m$^3$/s in Wadi Akhfi and 11.9 m$^3$/s in Wadi Elshlal (Table 2 and Figure 8), with high median values compared with Wadi El-Sheikh (east of the study area) during the flooding in October 1993 [42].

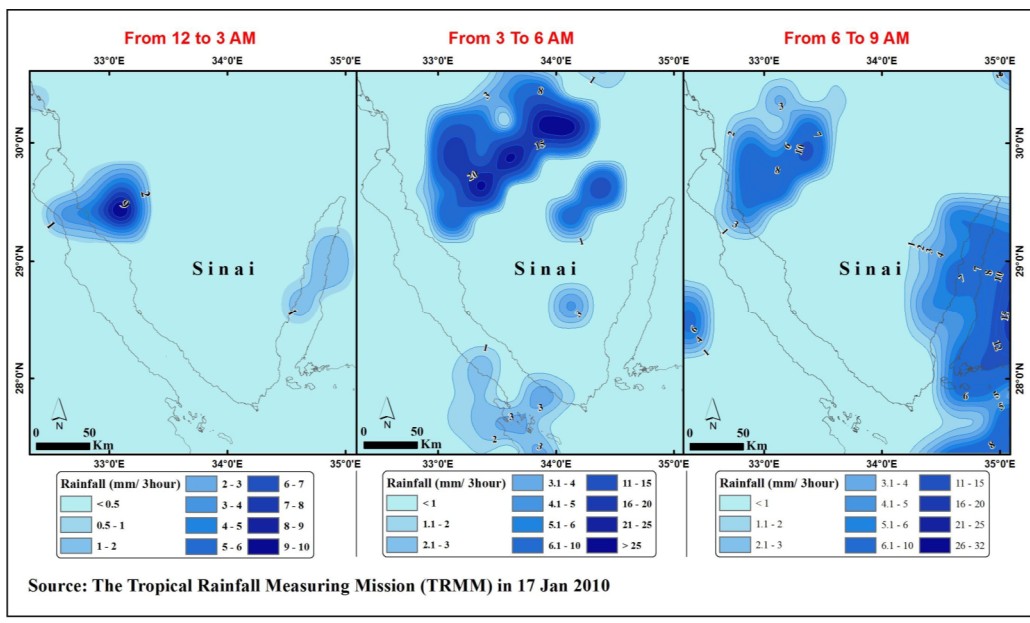

(a)

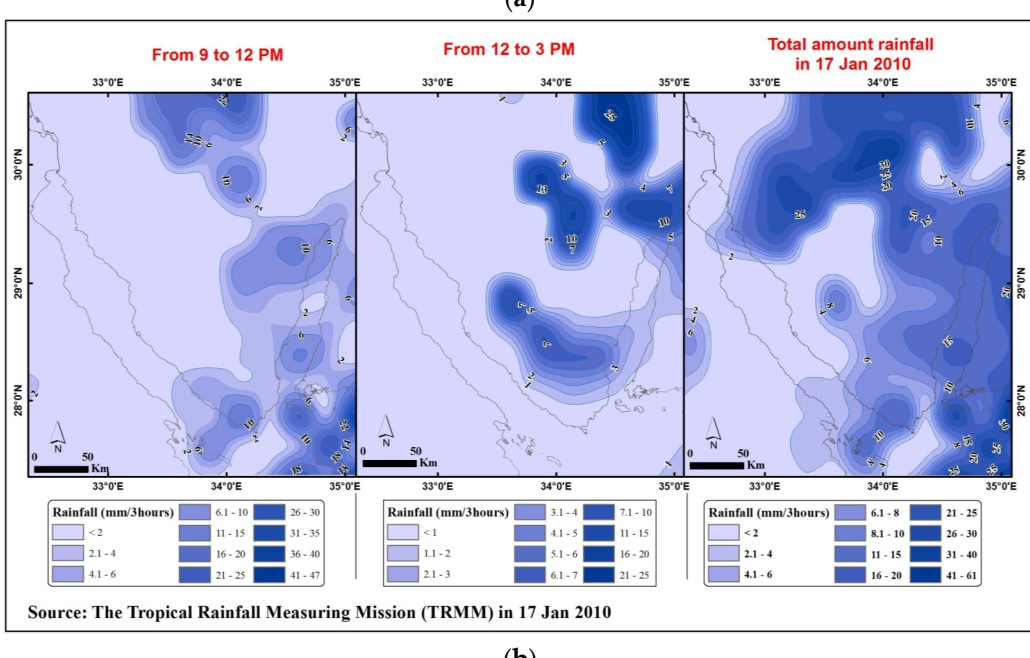

(b)

**Figure 4.** Shows the per 3 h TRMM accumulated rainfall data acquired on 17 January 2010: (**a**) the tropical rainfall measuring mission from 12 AM to 9 PM; (**b**) the tropical rainfall measuring mission from 9 AM to 3 PM and the total amount of rainfall on 17 Jan 2010.

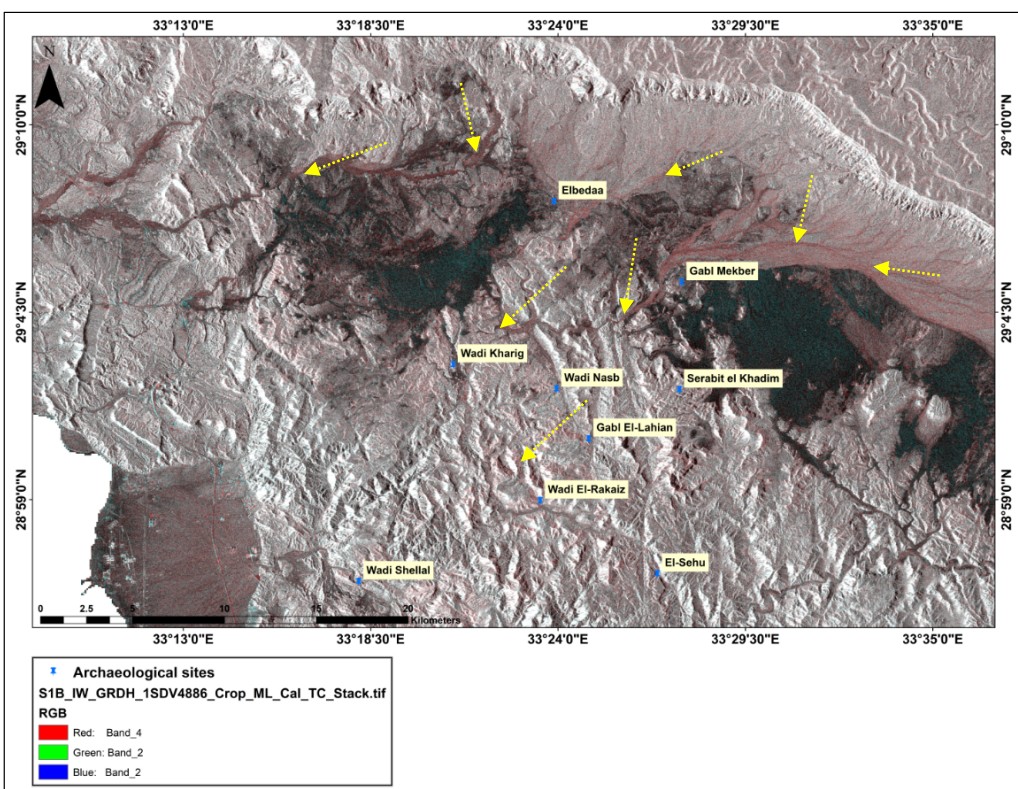

**Figure 5.** Flood hazard mapping presents the deepest area in the Wadi (in black) and the streams (in pink) from recent flooding denoted by the yellow arrows around the heritage sites using two Sentinel-1 image sets from 2017 and 2019.

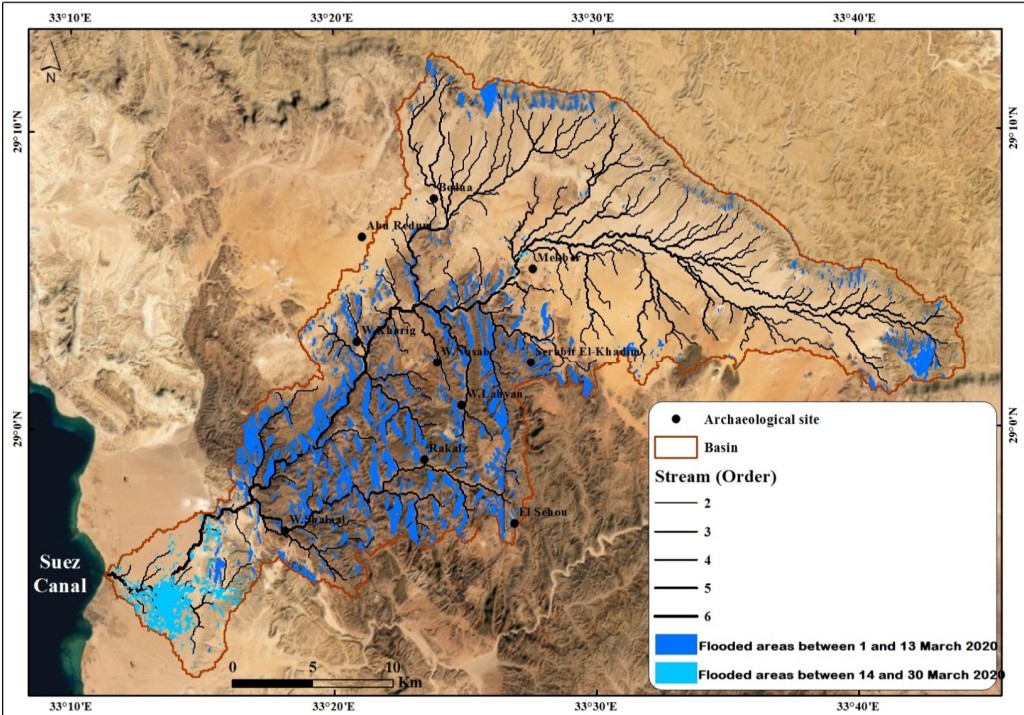

**Figure 6.** Threatened archaeological sites based on the extracted shapefile from Sentinel-1 data processed by GEE include the flooded regions between 1 and 13 March 2020 (in blue color), and flooded regions between 14 and 30 March 2020 (in sky blue color).

**Table 1.** Showing the methodology and results of the morphometric parameters for the study area.

| Serial Number | Morphometric Parameters | Methods | References | Serial Number | Value |
|---|---|---|---|---|---|
| Linear | Stream order (u) | Hierarchical order, DEM analyses by ArcGIS | Strahler (1964) | 1017 | |
| 1 | Sum of all stream numbers (SNu) | Counted from the analysis, (Nu is the number of order u) | | | |
| 2 | Stream length (Lu) Sum of all stream lengths (SLu) | DEM analyses by ArcGIS DEM analyses by ArcGIS | Horton (1945) | 1389.480530 | km km |
| 3 | Length of the main channel (Lm) | Measured through ArcGIS | - | 35.575907000 | km |
| 4 | Order of the main channel (K) | Identified from analysis | - | 6 | |
| 5 | Bifurcation ratio (Rb) | Rb = Nu/Nu + 1 | Horton (1945) | | |
| 6 | Weighted mean bifurcation ratio (WMRb) | WMRb = Sum {(Rb u/u + 1) 9 (Nu + Nu + 1)}/SumN | Strahler (1953) | 4.333136889 | |
| 7 | Sinuosity (Si) | Si = Lm/Lb | Gregory and Walling (1973) | 0.640557547 | |
| Relief 8 | Relief (R) | Calculated from DEM analyses | | 1398 | m |
| 9 | Relief ratio (Rr) | Rr = R/Lb (Schumm, 1956) | Schumm (1956) | 25.17151425 | |
| 10 | Ruggedness number (Rn) | Rn = D * R | Melton (1957) | 2691.251611 | |
| Areal 11 | Area (A) | Measured through ArcGIS | | 721.780815 | km$^2$ |
| 12 | Perimeter (P) | Measured through ArcGIS | | 248.330645 | km |
| 13 | Basin length (Lb) | Measured through ArcGIS | | 55.538971 | km |
| 14 | Drainage density (D) | D = SLu/A (Horton, 1945) | Horton (1945) | 1.925072683 | km$^{-1}$ |
| 15 | Stream frequency (F) | F = SNu/A (Horton, 1945) | Horton (1945) | 1.409015007 | km$^{-2}$ |
| 16 | Circulatory ratio (Rc) | Rc = 4 ∏ A/P2 | Miller (1953) | 0.147005756 | |
| 17 | Elongation ratio (Re) | Re = 2 (A/∏)0.5/Lb (Schumm, 1956) | Schumm (1956) | 0.122106146 | |
| 18 | Length of overland flow (Lo) | Lo = 1/(2D) | Horton (1945) | 0.259730453 | km |
| 19 | Drainage texture (Rt) | Rt = SNu/P | Horton (1945) | 4.095346348 | |
| 20 | Texture ratio (T) | T = SN1/P, where SN1 is the total number of 1st-order streams | Horton (1945) | 2.851037575 | |
| 21 | Form factor (Rf) | Rf = A/Lb2 | Horton ((1932) | 0.23399667 | |
| 22 | Basin shape index (Ish) | Ish = 1.27A/Lb2 | Hagget (1956) | 0.29717577 | |

**Table 2.** The hydrologic parameters in the four study area subcatchments.

| Parameter/Sub-Basins | Wadi Elseih | Wadi Akhfi | Wadi Elshlal | Wadi Naga Elfda |
|---|---|---|---|---|
| Length of main channel (m) | 40,133.7 | 19,684.8 | 24,107.3 | 35,575.9 |
| Slope % | 0.82 | 0.11 | 0.18 | 0.13 |
| Time of Concentration | 73.8 | 91.9 | 89.1 | 138.0 |
| M (Constant) | 0.6 | 0.6 | 0.6 | 0.6 |
| L | 40.13370079 | 19.68475132 | 24.1072743 | 35.5759073 |
| Slope (Percent) | 81.80358131 | 11.15246043 | 18.12121966 | 12.67380984 |
| lag time | 1.8 | 2.3 | 2.2 | 3.5 |
| Area (A) km$^2$ | 268.70 | 176.14 | 70.77 | 206.31 |
| Drainage Density (D) | 2.2 | 1.9 | 1.5 | 1.7 |
| Rain Depth (P) | 21.80 | 21.80 | 21.80 | 21.80 |
| Average of Slope (S) | 0.82 | 0.11 | 0.18 | 0.13 |
| Geology | 3.1 | 4.5 | 6.4 | 4.9 |
| R | 0.96 | 0.98 | 1.01 | 0.97 |
| Runoff Coefficient (C) | 1.54 | 5.06 | 2.78 | 4.19 |
| Runoff Volume (1000 m$^3$/h) | 86.6 | 190.4 | 43.2 | 183.5 |
| Peak flow rate m$^3$/s | 24.1 | 54.0 | 11.9 | 52.4 |

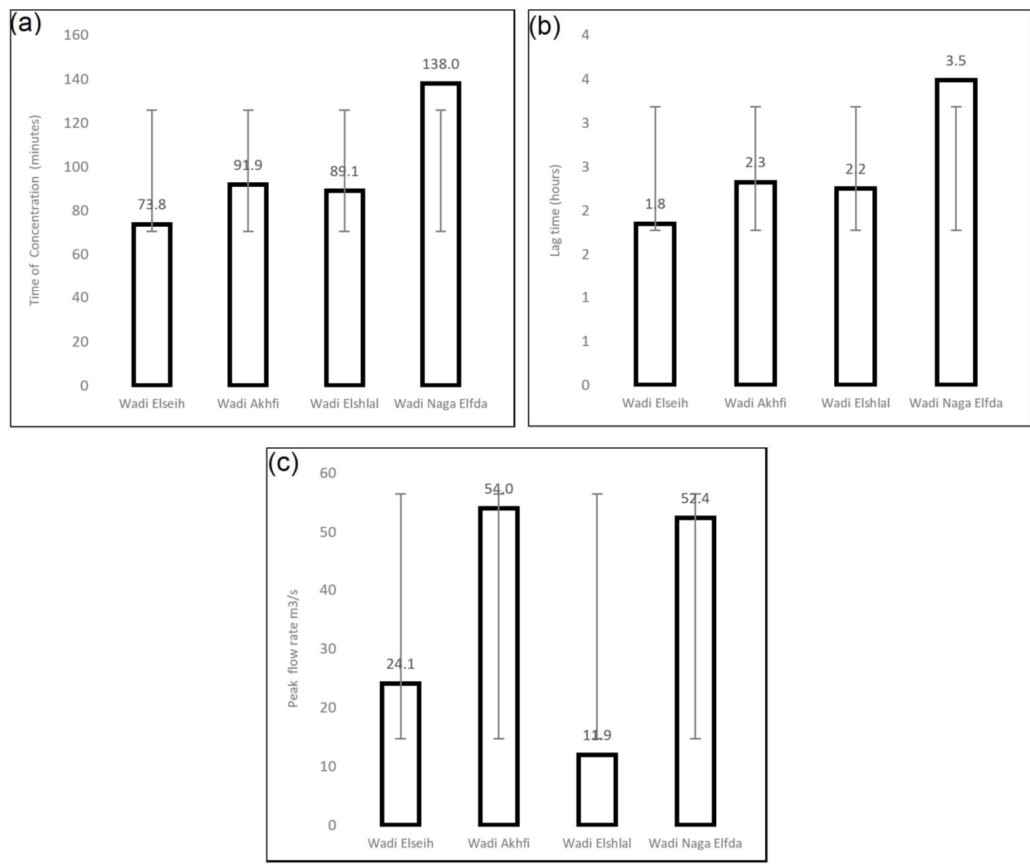

**Figure 7.** Charts of the hydrologic parameters of the four subcatchments: (**a**) time of concentration; (**b**) lag time; and (**c**) peak flow rate.

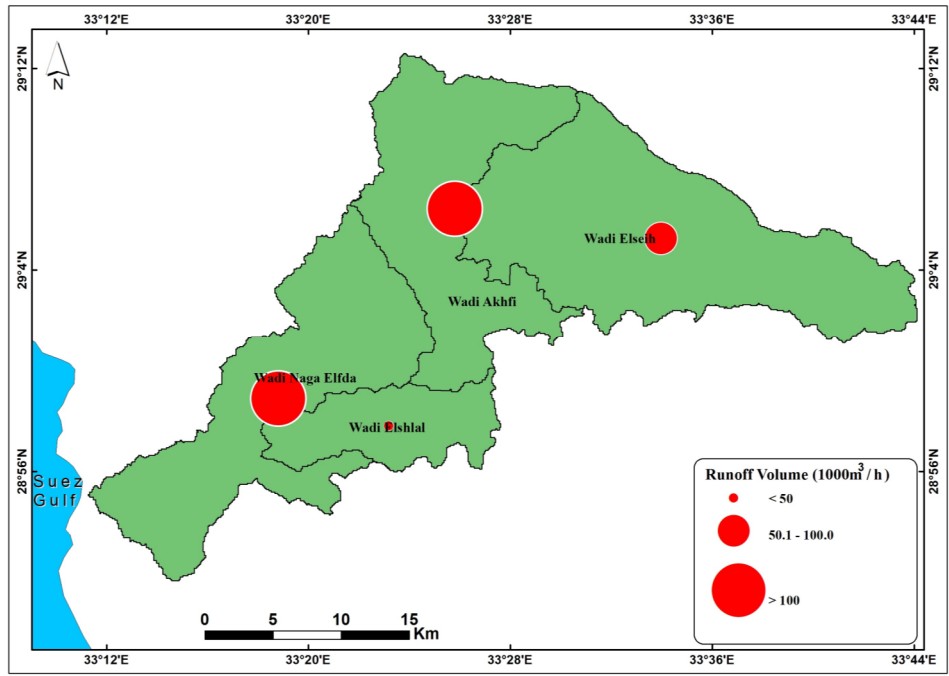

**Figure 8.** A map showing the runoff volume rate for the four subcatchments.

The flash flood discharge curve was estimated based on the methodologies proposed by [43] to derive a GIS-based spatially distributed unit hydrograph using an effective runoff coefficient of 33 mm [44]. Attwa et al. (2021) used this methodology to calculate

the total discharge from Al-Ambagi watershed in the Eastern desert [44]. We followed this method to compare the total discharge using this method and the one calculated using the TRMM data and assessed which method is more realistic. The total discharge resulting from the hypothetical 33 mm rainfall event was approximately 22,990,954.1 m$^3$ with a flow duration of 12 h; the maximum discharge was 4740.417 m$^3$/s and the time to peak discharge was 6 h (Figure 9), with high median values compared with the Gulf of Suez area (west of the study area) during the flood event of 17 January 2010 [45]. The results using both methods are different, likely because of the difference in the effective rainfall amount. Moreover, this variation is likely to be attributed to considering the watershed parameters (i.e., spatial variability in terms of geology and vegetation cover) in calculating the total discharge using the TRMM data. Flash floods are destructive to coastal areas, Ras Abu Rudeis city, infrastructure, and the precious archaeological sites in the study area. Although the four sub-basins show different vulnerability to flash flooding, the infrastructure and the archaeological sites require a well-designed management plan to be protected from these severe hazards. Floods can be controlled by several alternative methods such as planting vegetation to retain excess water, terrace slopes to reduce slope flow, building alluviums (manmade channels to divert water from flooding), and the construction of dykes, dams, reservoirs, or holding tanks to store extra water during flood periods. To manage the total runoff volume, we designed a model that takes into consideration nine parameters to select the optimum locations for dam construction for an economically effective management strategy. The proposed dams created according to the designed model for protecting the heritage sites are shown in Figure 10d, where the locations with nine values show the best location selected for dam construction.

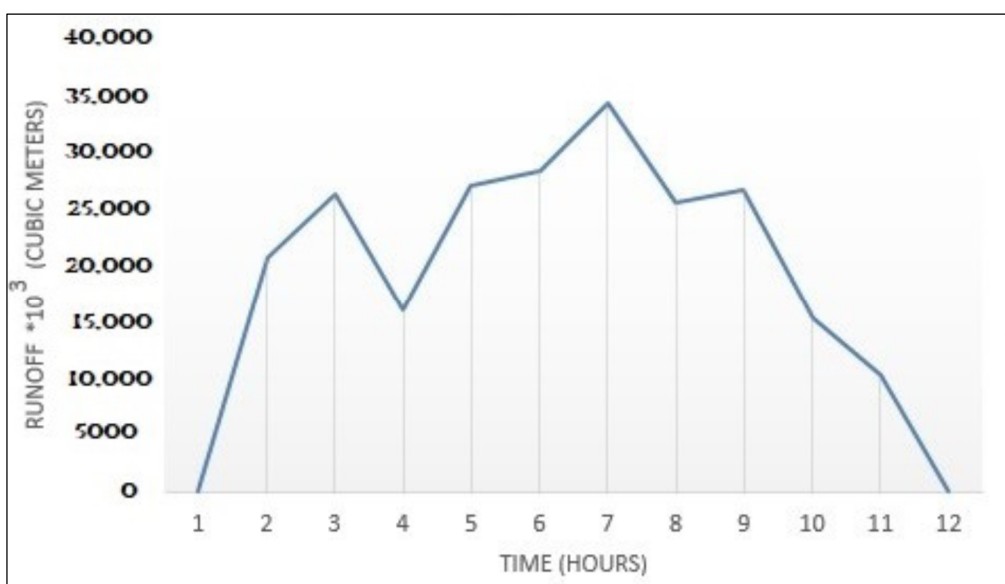

**Figure 9.** A hydrograph showing the discharge in the basin.

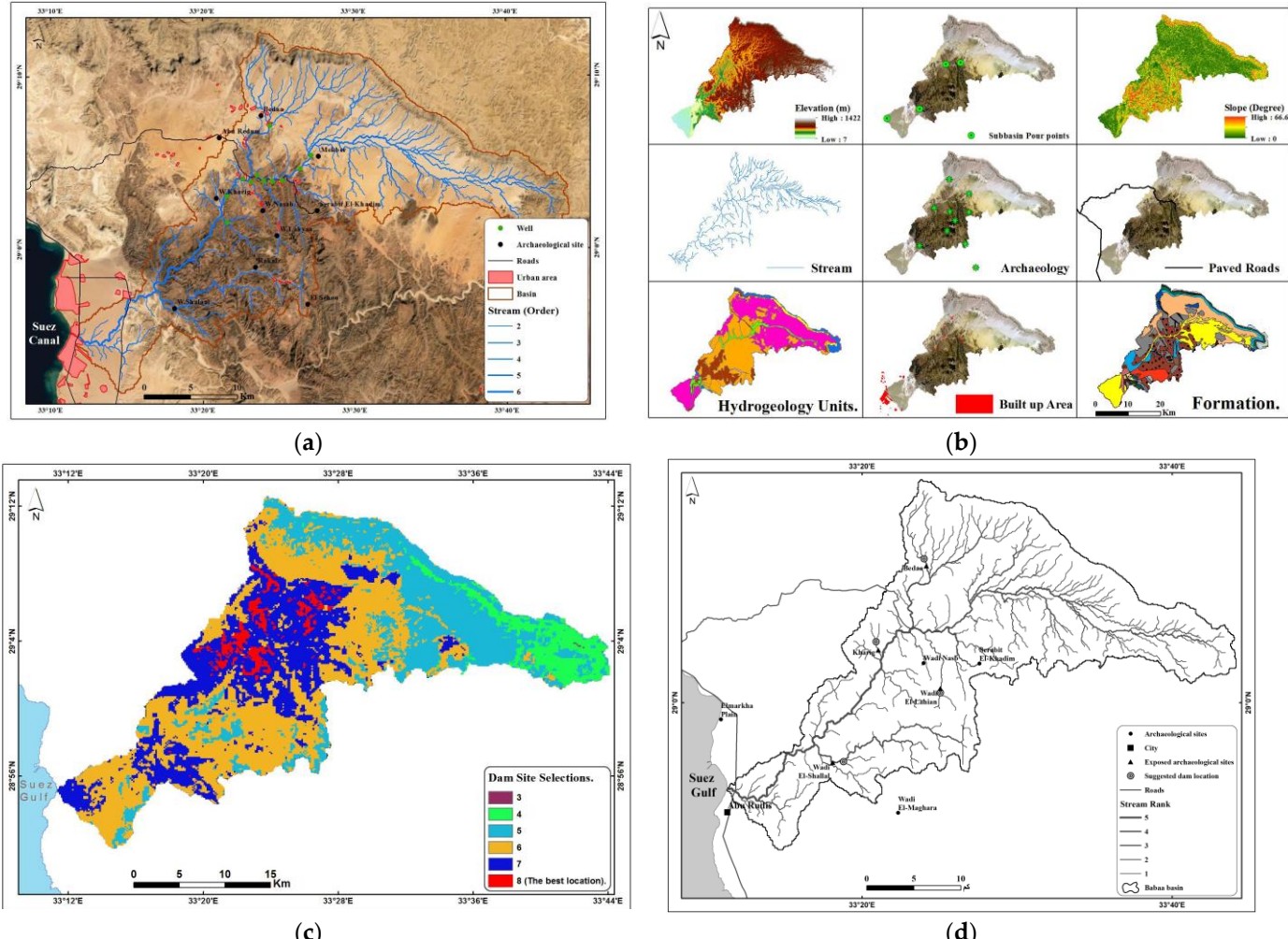

**Figure 10.** (**a**) Threatened areas according to the flood event and the proposed dams created according to the scientific model for protecting heritage sites based on the (**b**) nine parameters used in the (**c**) dam site selection model; (**d**) suggested dam sites in Babaa basin.

## 4. Recommendations

Due to the exposure of the study area to flood risk that could threaten its archaeological sites, built-up areas, and roads (Figure 10a), some dams were proposed based on the analysis of the outputs obtained from an innovative model. Nine factors were used for building that model: DEM, slope, streams, sub-basin pour points, archaeological sites, urban areas, roads, lithological units, and hydrogeological units. The proposed model was built as follows:

- The drainage network, elevation, slope, urban, archaeological sites, roads, sub-basin pour points, lithological units, and hydrogeological units were used as input.
- The lithological and hydrogeological units, having a field with numbers from one to nine equivalent to each unit, were converted to a raster image.
- The urban, archaeological sites, roads, sub-basin pour points, and stream layers were subjected to Euclidean distances for deriving the distance between the layer and surrounding grids.
- We prepared a suitability map by creating datasets that were reclassified into nine classes according to the dam construction's suitability to be ready for being combined where higher values are more suitable.
- Weighting and combining datasets: the reclassified datasets are ready to be combined and weighted according to importance (distance to streams = 19%, slope = 14%, dis-

tance to roads = 13%, hydrogeological units = 13%, lithological units = 13%, dem = 7%, distance to archaeological sites = 7%, distance to built-up area = 7%, distance to sub-basin pour points: 7%), where a parameter with a higher percentage has more influence as an input in the suitability model (Figure 10b–d).

## 5. Conclusions

Although Baba basin is an arid to semi-arid zone, it is strongly vulnerable to flooding because of the sudden rainfall and the characteristics of that basin. That site has great value because of the archaeological sites, oil fields, airports, resorts, and other infrastructure that are present in Abu Rudeis city (at the watershed outlet). These sites must be protected from the repeated damage caused by flash flood hazards. Remote sensing technology has become an irreplaceable Earth Observation (EO) tool over the last two decades and has recently been used for estimating rainfall intensity and monitoring iterative flooding events. In the same context, radar satellite data (e.g., Sentinel-1) have played an important role in detecting and evaluating the effects of flooding because of its reliability for collecting data on any climate situation 24 h a day. Moreover, GIS tools have received much interest from developers, which has enabled researchers to create innovative models based on optical and radar data. According to the integration between GIS techniques, remote sensing imagery (the TRMM accumulated rainfall data acquired on 17 January 2010, Sentinel-1 radar images acquired between 2017 and 2019, and Sentinel-1 data from March 2020), and depending on nine factors that consider effective parameters, a few optimum dam locations were recommended to mitigate the destructive effect of frequent flash floods. This model is a decision-making tool for detecting locations on the ground where multiple criteria overlap in geographic space. The main goal of this study was the development of a methodology to avoid and/or mitigate flash flood hazards at the study area by detecting suitable locations for dams' construction. The secondary goal was to identify the factors governing this selection. Such results can be beneficial for preserving archaeological sites around the world in terms of the ease of gathering information through satellites without cost or great effort. For future work, we recommend deploying another methodology called 2D Electrical Resistivity Tomography (ERT) at the optimum dam locations identified by the suitability model. This method is efficient in environmental studies. For instance, Attwa et al. (2021) integrated this method, remote sensing, and GIS for sustainably managing existing water resources in structurally controlled basins in the Eastern Desert [44]. Applying this method is expected to confirm the best dam locations and/or exclude some of the proposed dams, which may be economically efficient for decision-makers.

**Author Contributions:** W.A., D.R. and A.E. collected the data. Topographic maps and remote sensing imagery were analysed by W.A., D.R. and A.E. The manuscript was written by A.M.A.-H., A.M.M. and A.E. The article was revised by A.E. and R.L. All authors have read and agreed to the published version of the manuscript.

**Funding:** The research article received no external funding as the research activities were performed at the authors' research centers.

**Institutional Review Board Statement:** Not applicable.

**Informed Consent Statement:** Not applicable.

**Data Availability Statement:** Not applicable.

**Acknowledgments:** The authors would like to thank the National Authority for Remote Sensing and Space Science (NARSS) in Cairo, Egypt, and the National Research Centre, Cairo, Egypt, Faculty of Arts, Cairo University, Egypt, and Faculty of Arts, Banha University, Egypt for supporting their research activities. Special thanks go to the Italian National Research Council, C.da Santa Loja, Tito Scalo, Potenza, Italy for supporting the research with some required data.

**Conflicts of Interest:** The authors declare no conflict of interest.

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
