# Peer review of "On the Use of Radar and Optical Satellite Imagery for the Monitoring of Flood Hazards on Heritage Sites in Southern Sinai, Egypt"

_sustainability, doi:10.3390/su14095500_

Round 1

Reviewer 1 Report

The manuscript “On the use of Radar and optical satellite imagery for the monitoring of flood hazard on the heritage sites in southern sinai, egypt” by Wael et al. disclosed the exposure of heritage in wadi baba by combining hydrology model and RS images.

I suspect that this article does not have real results. Just a qualitative analysis of the impact of the flood derived from TRMM.

Line 47-48 Radar satellite data can not use mapping of flood volume, can use mapping of flood extent.

Line 52 this paragraph can not express the intention of this article. TRMM, SRTM are data sources, ARCSWAT, arcgis, Blue kenue software, SNAP are the method tools. However, how to evaluate the flood risk is not clear.

Line 94 this paragraph should be moved to line 98(section methods)

Line 124 SRTM is DEM derived from SAR product. SRTM is not SAR data and should be divided as DEM parameter.

Line 171 section 3.1 I can not get the point about satellite images results, what result is it?

Line 196 section 3.2 this section is more likely method.

Line 203-213 did runoff volume come from observation or others? I suggest this paragraph should move to 2.1 data collection.

Line 215-224 are this result derived form ARCSWAT? Table 2 and Figure 5 show the Parameters, where is the result of ARCSWAT?

Author Response

Comments and Suggestions for Authors

The manuscript “On the use of Radar and optical satellite imagery for the monitoring of flood hazard on the heritage sites in southern Sinai, Egypt” by Wael et al. disclosed the exposure of heritage in wadi baba by combining hydrology model and RS images.

I suspect that this article does not have real results. Just a qualitative analysis of the impact of the flood derived from TRMM.

Authors: first of all, the Tropical Rainfall Monitoring Mission (TRMM ) data and the measurements from meteorological stations of Abu Zinema are real data, also, the radar data were automatically processed (software and platform) and the extracted data are surely real. Also, there are many published articles that depended on the same input data (e.g. (Gabr, S. and El Bastawesy, M., 2015. Estimating the flash flood quantitative parameters affecting the oil-fields infrastructures in Ras Sudr, Sinai, Egypt, during the January 2010 event. The Egyptian Journal of Remote Sensing and Space Science, 18(2), pp.137-149.) Referred before to the south Sinai area with the same problem (flood events) effects.  Even the recommended solutions are totally built on the remote sensing data.

 Line 47-48 Radar satellite data cannot use mapping of flood volume, can use mapping of flood extent.

Authors: the word volume is replaced with extent based on the reviewer’s comment

Line 52 this paragraph can not express the intention of this article. TRMM, SRTM are data sources, ARCSWAT, arcgis, Blue kenue software, SNAP are the method tools. However, how to evaluate the flood risk is not clear.

Authors: the paragraph has been modified (please check the paragraph lines 55-62.

Line 94 this paragraph should be moved to line 98 (section methods)

Authors:  the paragraph has been moved to the method section based on the reviewer comment (lines; 98-100).

Line 124 SRTM is DEM derived from SAR product. SRTM is not SAR data and should be divided as DEM parameter.

Authors:  In our case, this data was extracted from the Sentinel-1 data in internal process in SNAP software as step in the pre-processing steps in the radar data. Please check (Jo, M. J; Osmanoglu, B; Zhang, B; Wdowinski, S. Flood extent mapping using dual-polarimetric Sentinel-1 synthetic aperture radar imagery. Int. Arch. Photogramm. Remote Sens. Spat. Inf. Sci 42, 2018; 3; pp.711-713.)

Line 171 section 3.1 I can not get the point about satellite images results, what result is it?

Authors: we added new radar data processed by Google Earth Engine (GEE) for showing more about the results of the flooding events on the archaeological sites of the study area, and the specified in March 2020 (please check sections; (2.2.3. Radar data was analysed using the GEE platform, also. 3.1. Satellite images results).

Line 196 section 3.2 this section is more likely method.

Authors: this section has been totally modified and additional details and figures have been added (please check lines 242-302

Line 203-213 did runoff volume come from observation or others? I suggest this paragraph should move to 2.1 data collection.

Authors: runoff volume has been extracted and calculated based on the formula (4), and the rainfall amount has been extracted from the TRMM data based on Abu Zenima observation station. And, the authors discussed this point in section 3.2. Extracting the morphometric and Hydrologic parameters and conclusion section.

Line 215-224 are this result derived form ARCSWAT? Table 2 and Figure 5 show the Parameters, where is the result of ARCSWAT?

Authors: The term ARCSWAT has been removed and replaced with ARC hydro tool for being more accurate based on the reviewer’s comment.

Reviewer 2 Report

Dear Editor.

I have finished my review on the proposed paper “On the Use of Radar and Optical Satellite Imagery for the Monitoring of Flood Hazard on the Heritage Sites in Southern Sinai, Egypt” sustainability-1613311-peer-review-v1.

Summary of the manuscript:

In the proposed paper, the authors’ goal is to predict the maximum flood discharge and the respective flood zones in the heritage sites of Egypt in southern Sinai Peninsula. According to the authors results, the study area is exposed to flood risk that significantly threatened the heritage sites.

General review:

  1. Generally, the manuscript presents a very interesting topic and the specific research seems to include some significant points for the research community of this field.
  2. The proposed paper is very well written with very good use of English language. Except some minor grammatical mistakes and word errors, this paper is written with a very good scientific style. The authors should check again the paper to correct these minor mistakes.
  3. The proposed paper is very well structured. It begins with the Introduction, in which there is an effort to provide previous studies with similar scientific content. However, there are very few references that provide respective studies from other countries or/and other studies that deal with extreme rainfall and flood events in the broader area. More literature should be added in the Introduction. Below I give some papers to add and begin with. At the end of Introduction, authors clearly state the goals of the research.
  4. The methodology is generally very interesting. However, some significant parts are missing. Below I give specific comments.
  5. The results are confusing. Below I give specific comments.

6.There is no Discussion of the results. See below specific comments.

  1. Conclusions are appropriate for this paper.

Points for revision:

Lines 36-37: Please, provide the economic losses in Euros or American Dollars.

Line 37: {….Egyptian pounds” [3].} Here, you have added right quotation (”), but I didn’t find where is the left quotation (“).

Lines 46-51: In these lines you say about the utilization of satellite data in flood mapping etc. However, there are some very recent studies that explored the applicability of satellite-based precipitation products for event-based hydrological modelling of flash floods in various geomorphology ungauged catchments (Sapountzis et al. 2021, Gilewski and Nawalany, 2018). You should add a phrase about this in your Introduction adding the proposed literature.

Line 91: “….and rainfall ground data….”. What you mean? Is there in the study area an operating rain gauge? Where is the location. What is the time step? Give some information.

GENERALLY: Which are the dates of the rainfall events that you have investigate?

Lines 137, 143, 152, 159: Here, you used some equations for time of concentration, lag time, runoff volume and the maximum flow. However, you didn’t explain why you chose the specific equation.

Figure 3: Please, provide the figure with better resolution.

Lines 172-173: Ok! Here, in results you informed the readers about the date of flood event (???) (17 January 2010). You should provide the rain graphs both from TRMM and the rain gauge.

Lines 208-214 and table 2: Here, you provide some numbers about the Runoff Volume (1000 m3/ hour) and the Peak flow rate m3/s. However, these numbers are confusing.

  1. To calculate the Peak flow rate you simple divided the Runoff Volume with 3600 seconds (Wadi Elseih for example: 86600/3600=24.1). But equation (5) does not use the Runoff Volume (V) that you calculated with equation (4).
  2. Also, the values of Peak flow rate are very low, almost negligible. For example, the specific discharge (m3/s*km2) for all your study area is 1436.49/721.78=1.99 m3/s*km2. According to previous studies values of specific discharge in a range between 9–11 m3/s*km2 are common for extreme flood events (Gaume et al., 2009; Marchi et al., 2009). Please, explain why these values are so low.
  3. Provide the flood hydrographs for the investigated flood events. Also, I do not understand why you used the TRMM if you have ground rainfall data.

Line 220: And here (at the end of the paper) you say about an “hypothetical rainfall of 10mm”. But you said that you investigate specific flood events. I am very confused….

DISCUSSION: There is no discussion of the results. There is the section 3, however, discussion within the context of comparing the results of the paper with other studies, does not exist. I searched the paper from, but I did not find not a single reference. You should compare your results with previously published studies. 

You have used very much self-citations.

References

Gaume, E., Bain, V., Bernardara, P., Newinger, O., Barbuc, M., Bateman, A.,… Viglione, A. (2009). A compilation of data on European flash floods. Journal of Hydrology, 367(1), 70–78. https://doi.org/10.1016/j.jhydrol.2008.12.028.

Gilewski P. and Nawalany M. (2018). Inter-comparison of rain-gauge, radar, and satellite (IMERG GPM) precipitation estimates performance for rainfall-runoff modeling in a mountainous catchment in Poland. Water, 10, 1665. https://doi.org/10.3390/w10111665.

Sapountzis M., Kastridis A., Kazamias A-P., Karagiannidis A., Nikopoulos P., and Lagouvardos K. 2021. Utilization and uncertainties of satellite precipitation data in flash flood hydrological analysis in ungauged watersheds. Global NEST Journal, 23(3), 388-399. doi.org/10.30955/gnj.003905

Marchi, L., Borga, M., Preciso, E., Sangati, M., Gaume, E., Bain, V., & Pogacˇnik, N. (2009). Comprehensive post-event survey of a flash flood in Western Slovenia: Observation strategy and lessons learned. Hydrological Processes, 23(26), 3761–3770.

Author Response

Dear Editor.

I have finished my review on the proposed paper “On the Use of Radar and Optical Satellite Imagery for the Monitoring of Flood Hazard on the Heritage Sites in Southern Sinai, Egypt” sustainability-1613311-peer-review-v1.

Summary of the manuscript:

In the proposed paper, the authors’ goal is to predict the maximum flood discharge and the respective flood zones in the heritage sites of Egypt in southern Sinai Peninsula. According to the authors results, the study area is exposed to flood risk that significantly threatened the heritage sites.

General review:

  1. Generally, the manuscript presents a very interesting topic and the specific research seems to include some significant points for the research community of this field.
  2. The proposed paper is very well written with very good use of English language. Except some minor grammatical mistakes and word errors, this paper is written with a very good scientific style. The authors should check again the paper to correct these minor mistakes.

Authors:  the English of the whole article body has been revised

  1. The proposed paper is very well structured. It begins with the Introduction, in which there is an effort to provide previous studies with similar scientific content. However, there are very few references that provide respective studies from other countries or/and other studies that deal with extreme rainfall and flood events in the broader area. More literature should be added in the Introduction. Below I give some papers to add and begin with. At the end of Introduction, authors clearly state the goals of the research.

Authors:  All the recommended references have been added in both Introduction and discussion sections.

  1. The methodology is generally very interesting. However, some significant parts are missing. Below I give specific comments.

Authors:  all the recommended points have been considered and totally done.

  1. The results are confusing. Below I give specific comments.

Authors:  new details and figures have been added based on the reviewer’s comments

  1. There is no Discussion of the results. See below specific comments.

Authors:  additional texts and figures have been added to the section please check lines; 188-302

  1. Conclusions are appropriate for this paper.

Points for revision:

Lines 36-37: Please, provide the economic losses in Euros or American Dollars.

Authors: the required loss value has been added in Euro (please check lines 37-38).

Line 37: {….Egyptian pounds” [3].} Here, you have added right quotation (”), but I didn’t find where is the left quotation (“).

Authors: The quotation has been deleted based on the reviewer’s comment

Lines 46-51: In these lines you say about the utilization of satellite data in flood mapping etc. However, there are some very recent studies that explored the applicability of satellite-based precipitation products for event-based hydrological modelling of flash floods in various geomorphology ungauged catchments (Sapountzis et al. 2021, Gilewski and Nawalany, 2018). You should add a phrase about this in your Introduction adding the proposed literature.

Authors: All the recommended references have been added.

Line 91: “….and rainfall ground data….”. What you mean? Is there in the study area an operating rain gauge? Where is the location. What is the time step? Give some information.

Authors: The required details have been added please check lines 93-94.

GENERALLY: Which are the dates of the rainfall events that you have investigate?

Authors: The dates of the rainfalls have been added (17 Jan2010 and March 2020), in the same context, new details and figures have been added please check the data and methods, the results, and discussion sections.

Lines 137, 143, 152, 159: Here, you used some equations for time of concentration, lag time, runoff volume and the maximum flow. However, you didn’t explain why you chose the specific equation.

Authors: Because these equations are more suitable for the Arid/Semi-arid zones such as the desert of Saudi Arabia (please check. Al-Amri, N.S., Ewea, H.A. and Elfeki, A.M., 2022. Revisit the rational method for flood estimation in the Saudi arid environment. Arabian Journal of Geosciences, 15(6), pp.1-14.)

Figure 3: Please, provide the figure with better resolution.

Authors: All the figures have been repaired with higher resolution

Lines 172-173: Ok! Here, in results you informed the readers about the date of flood event (???) (17 January 2010). You should provide the rain graphs both from TRMM and the rain gauge.

Authors: the hydrograph and calculated discharge for the flooding event have been added please check figures4 and 9. Also, new details have been added. 

Lines 208-214 and table 2: Here, you provide some numbers about the Runoff Volume (1000 m3/ hour) and the Peak flow rate m3/s. However, these numbers are confusing.

Authors: runoff volume explains the total amount of the runoff that accumulated from rainfall (fig. 6 and table 2), while the peak flow shows the maximum rate of runoff volume (fig. 5c).

  1. To calculate the Peak flow rate you simple divided the Runoff Volume with 3600 seconds (Wadi Elseih for example: 86600/3600=24.1). But equation (5) does not use the Runoff Volume (V) that you calculated with equation (4).

Authors: the peak flow equation (5)  is dependent on, Q = peak flow rate (maximum runoff), m 3/s A =catchment area, I = rainfall intensity, mm/hour, C = runoff coefficient, and did not use runoff volume. Where the runoff coefficient is the most important factor in the runoff volume equation that depended on the rational method equations. While equation 4 is used for calculating the runoff volume.

  1. Also, the values of Peak flow rate are very low, almost negligible. For example, the specific discharge (m3/s*km2) for all your study area is 1436.49/721.78=1.99 m3/s*km2. According to previous studies values of specific discharge in a range between 9–11 m3/s*km2 are common for extreme flood events (Gaume et al., 2009; Marchi et al., 2009). Please, explain why these values are so low.

Author: the Peak flow value is calculated by m3/second, in our study area (Arid zone) the values 24.1, 54.0, 11.9, 52.4 m3/second for the studies wadies are acceptable, also the total discharge calculated value is 22990954.1 cubic meters with a flow duration of twelve hours which considers normal value in the studied area. 

  1. Provide the flood hydrographs for the investigated flood events. Also, I do not understand why you used the TRMM if you have ground rainfall data.

Authors: the flood hydrograph is added please check (fig.9, line 302), the TRMM data were used to display our results more accurate for the amount of Rainfall in all the basins because the Abu Znima station is located at the coastline of the Red Sea, its location maybe will affect the accurate result of the rainfall, while most of our study area is located in a different topographic situation (with higher elevation and differ climate situation).

Line 220: And here (at the end of the paper) you say about an “hypothetical rainfall of 10mm”. But you said that you investigate specific flood events. I am very confused….

Author: The hydrograph of the discharge is recalculated to 33mm instead of 10mm based on the reviewer’s request please check line 276.  

DISCUSSION: There is no discussion of the results. There is the section 3, however, discussion within the context of comparing the results of the paper with other studies, does not exist. I searched the paper from, but I did not find not a single reference. You should compare your results with previously published studies.

 Authors: new details, data, and references in the results and discussion section have been added please check lines 189-304.

You have used very much self-citations.

  1. Lasaponara, R; Murgante, B; Elfadaly, A; Qelichi, M; Shahraki, S; Wafa, O; Attia, W. Spatial open data for monitoring risks and preserving archaeological areas and landscape; Case studies at Kom el Shoqafa, Egypt and Shush, Iran. Sustainability 9, 2017; 4; p.572.
  2. Elfadaly, A; Lasaponara, R; Murgante, B; Qelichi, M. M. 2017, July; Cultural heritage management using analysis of satellite images and advanced GIS techniques at East Luxor, Egypt and Kangavar, Iran (a comparison case study). In International Conference on Computational Science and Its Applications, pp. 152-168. Springer, Cham, 2017.
  3. Elfadaly, A; Lasaponara, R. On the use of satellite imagery and GIS tools to detect and characterize the urbanization around heritage sites; the case studies of the Catacombs of Mustafa Kamel in Alexandria, Egypt and the Aragonese Castle in Baia, Italy. Sustainability 11, 2019; 7; p.2110.
  4. Elfadaly, A; Murgante, B; Qelichi, M. M; Lasaponara, R; Hosseini, A. 2019, July; A Comparative Analysis of Temporal Changes in Urban Land Use Resorting to Advanced Remote Sensing and GIS in Karaj, Iran and Luxor, Egypt. In International Conference on Computational Science and Its Applications, pp. 689-703. Springer, Cham.
  5. Elfadaly, A; Lasaponara, R; Murgante, B; Qelichi, M. M. 2017, July; Cultural heritage management using analysis of satellite images and advanced GIS techniques at East Luxor, Egypt and Kangavar, Iran (a comparison case study). In International Conference on Computational Science and Its Applications, pp. 152-168. Springer, Cham, 2017.

References

Gaume, E., Bain, V., Bernardara, P., Newinger, O., Barbuc, M., Bateman, A.,… Viglione, A. (2009). A compilation of data on European flash floods. Journal of Hydrology, 367(1), 70–78. https://doi.org/10.1016/j.jhydrol.2008.12.028.

Authors: Instead it the authors used (Llasat, M.C., Llasat-Botija, M., Prat, M.A., Porcu, F., Price, C., Mugnai, A., Lagouvardos, K., Kotroni, V., Katsanos, D., Michaelides, S. and Yair, Y., 2010. High-impact floods and flash floods in Mediterranean countries: the FLASH preliminary database. Advances in Geosciences23, pp.47-55.), because the recommended one is by purchase

 Gilewski P. and Nawalany M. (2018). Inter-comparison of rain-gauge, radar, and satellite (IMERG GPM) precipitation estimates performance for rainfall-runoff modeling in a mountainous catchment in Poland. Water, 10, 1665. https://doi.org/10.3390/w10111665.

Authors: the recommended reference has been added

Sapountzis M., Kastridis A., Kazamias A-P., Karagiannidis A., Nikopoulos P., and Lagouvardos K. 2021. Utilization and uncertainties of satellite precipitation data in flash flood hydrological analysis in ungauged watersheds. Global NEST Journal, 23(3), 388-399. doi.org/10.30955/gnj.003905

Authors: the recommended reference has been added

Marchi, L., Borga, M., Preciso, E., Sangati, M., Gaume, E., Bain, V., & Pogacˇnik, N. (2009). Comprehensive post-event survey of a flash flood in Western Slovenia: Observation strategy and lessons learned. Hydrological Processes, 23(26), 3761–3770.

Authors: the recommended reference has been added

Round 2

Reviewer 1 Report

The paper has undergone principal revisions and improvements since the last submission. 

Why did authors change arcswat to ARChydro? ARCSWAT and ARCHydro are different tools.

Author Response

Reviewer 1

Comments and Suggestions for Authors

The paper has undergone principal revisions and improvements since the last submission. 

Why did authors change arcswat to ARChydro? ARCSWAT and ARCHydro are different tools.

Authors: Based on the SRTM data processing, the ArcHydro model gave us the opportunity to widen our results to cover the study requirements more than the ArcSWAT tool

Reviewer 2 Report

Dear authors.

Thank you for the provided responses. I think that the the paper has significantly improved. However, I have three additional comments.

  1. About the Specific discharge (m3/s*km2). Your response was:

"the Peak flow value is calculated by m3/second, in our study area (Arid zone) the values 24.1, 54.0, 11.9, 52.4 m3/second for the studies wadies are acceptable, also the total discharge calculated value is 22990954.1 cubic meters with a flow duration of twelve hours which considers normal value in the studied area." 

However, this response does not answer to my concerns. The Specific discharge (m3/s*km2) is very common in Hydrology and widely used when we want to compare the peak flows from different watersheds. We just divide the peak flow (m3/s) with the watershed area (km2). 

So, the specific discharge of your study area is negligible compared with previous studies in Mediterranean. And I do not understand how is possible, this low values of peak flow, to give devastating flood events. You should provide literature from previous studies, showing flood events with so low peak flows, to support your response. 

2. In Introduction, add three more references to enhance the quality of this section.In lines 45-46, with the reference [7], also add Kundzewicz 2009. In lines 48-49, with the reference [9], also add Thomson et al. 2003. In lines 50-51, with the reference [10], also add Tzoutzios and Kastridis 2020.  

Kundzewicz, Z.W. Non-structural flood protection and sustainability, International Water Resources Association. Water Int. 2009, 27, 3–13.

Thompson, R.; Humphrey, J.; Harmer, R.; Ferris, R. Restoration of Native Woodland on Ancient Woodland Sites, Forestry Commission Practice Guide; Forestry Commission: Edinburgh, UK, 2003.

Tzioutzios C., Kastridis A. Multi-Criteria Evaluation (MCE) Method for the Management of Woodland Plantations in Floodplain Areas. ISPRS Int. J. Geo-Inf. 2020, 9, 725. https://doi.org/10.3390/ijgi9120725

3. In lines 40-41 you forgot to change the Egyptian pounds to Euros.

Author Response

Reviewer 2

Comments and Suggestions for Authors

Dear authors.

Thank you for the provided responses. I think that the paper has significantly improved. However, I have three additional comments.

  1. About the Specific discharge (m3/s*km2). Your response was:

"the Peak flow value is calculated by m3/second, in our study area (Arid zone) the values 24.1, 54.0, 11.9, 52.4 m3/second for the studies wadies are acceptable, also the total discharge calculated value is 22990954.1 cubic meters with a flow duration of twelve hours which considers normal value in the studied area." 

However, this response does not answer to my concerns. The Specific discharge (m3/s*km2) is very common in Hydrology and widely used when we want to compare the peak flows from different watersheds. We just divide the peak flow (m3/s) with the watershed area (km2). 

So, the specific discharge of your study area is negligible compared with previous studies in Mediterranean. And I do not understand how is possible, this low values of peak flow, to give devastating flood events. You should provide literature from previous studies, showing flood events with so low peak flows, to support your response. 

Authors: the required information have been added with the citations (please check the lines; 264-266 and 283-285.

  1. In Introduction, add three more references to enhance the quality of this section. In lines 45-46, with the reference [7], also add Kundzewicz 2009. In lines 48-49, with the reference [9], also add Thomson et al. 2003. In lines 50-51, with the reference [10], also add Tzoutzios and Kastridis 2020.  

Kundzewicz, Z.W. Non-structural flood protection and sustainability, International Water Resources Association. Water Int. 2009, 27, 3–13.

Authors: the recommended reference has been added in the text based on the reviewer comment.

Thompson, R.; Humphrey, J.; Harmer, R.; Ferris, R. Restoration of Native Woodland on Ancient Woodland Sites, Forestry Commission Practice Guide; Forestry Commission: Edinburgh, UK, 2003.

Authors: the recommended reference has been added in the text based on the reviewer comment.

Tzioutzios C., Kastridis A. Multi-Criteria Evaluation (MCE) Method for the Management of Woodland Plantations in Floodplain Areas. ISPRS Int. J. Geo-Inf. 2020, 9, 725. https://doi.org/10.3390/ijgi9120725

Authors: the recommended reference has been added in the text based on the reviewer's comment.

  1. In lines 40-41 you forgot to change the Egyptian pounds to Euros.

Authors: The Egyptian pounds have been changed to Euros based on the reviewer's comment please check line 40

Round 3

Reviewer 2 Report

Dear authors.

Thank you for your effort to address my comments.